



# Attributing the occurrence and intensity of extreme events with the flow analogues method

Robin Noyelle[1,2], Davide Faranda[1,3,4], Yoann Robin[1], Mathieu Vrac[1], and Pascal Yiou[1]

[1]Laboratoire des Sciences du Climat et de l'Environnement, UMR 8212 CEA-CNRS-UVSQ, Université Paris-Saclay & IPSL, Gif-sur-Yvette, 91191, France
[2]Institute for Atmospheric and Climate Science, ETH Zürich, Zürich, Switzerland
[3]London Mathematical Laboratory, 8 Margravine Gardens London, W6 8RH, London, United Kingdom
[4]Laboratoire de Météorologie Dynamique/IPSL, École Normale Supérieure, PSL Research University, Sorbonne Université, École Polytechnique, IP Paris, CNRS, 75005, Paris, France

**Correspondence:** Robin Noyelle (robin.noyelle@lsce.ipsl.fr)

**Abstract.**

Extreme event attribution methodologies have been proposed to estimate the impacts of anthropogenic global warming on observed climatological and meteorological extremes. The classical risk-based approach uses Extreme Value Theory (EVT) to derive changes in the unconditional probabilities of yearly maxima but bears the risk of comparing events with different

dynamical mechanisms. The flow analogues method on the other hand is a conditional attribution method which compares events with similar synoptic scale dynamics. Here we propose a procedure for estimating both the intensity change and the probability ratio of observed extreme events with this method. We illustrate the procedure on three recent extreme events in Europe and compare the results obtained to the EVT-based approach. We show that the conditional flow analogues method gives more significant results for these events, which suggests a stronger climate change signal than the one detected with the

unconditional approach.

## 1 Introduction

Extreme meteorological and climatological events affect negatively societies and ecosystems (Clarke et al., 2022). The frequency and intensity of these events can change under anthropogenic global warming, further exacerbating their impacts (Seneviratne et al., 2021). The occurrence of extreme events with strong societal impacts has sparkled the development of

15 so-called extreme events attribution methods which aim is to assess the role of anthropogenic global warming (AGW) in the occurrence and intensity of these extremes. The idea of risk-based extreme event attribution methods (National Academies of Sciences Engineering and Medicine, 2016) is to compare the probabilities $\mathbb{P}(X \geq x \mid F)$ of an observable $X$ exceeding a certain *observed* level $x$ during an extreme event in a counterfactual world ($F = 0$) and in a factual world ($F = 1$). The difference between the two worlds usually lies in the anthropogenic influence on the climate, often measured in terms of increases

in the global or regional mean temperature (GMST or RMST). GMST or RMST indeed integrate the effects of multiple anthropogenic forcings and, at least up to the recent years, extremes have mostly responded linearly to the increase in GMST or





RMST (Arnell et al., 2019; Van Loon and Thompson, 2023). In extreme events attribution, the factual world refers to the current state of the climate, which includes the influence of human activities, such as greenhouse gas emissions, land use changes, and other anthropogenic factors that have contributed to global warming and climate change. In contrast, the counterfactual world

is a hypothetical scenario that represents what the climate would be without these human influences, essentially reflecting a pre-industrial or natural climate baseline. If $\mathbb{P}(X \geq x \mid F = 1) > \mathbb{P}(X \geq x \mid F = 0)$ then it is more likely to observe an event with the intensity $x$ in the factual world, and it is thus inferred that anthropogenic climate change made the observed event more likely. One typically reports the ratio between these probabilities (called the *probability ratio*), which gives how more (or less) likely is the event in the factual world compared to the counterfactual world (Stott et al., 2016).

This framework requires the estimation of the probabilities of the observable $X$ reaching the level $x$ in the counterfactual and factual worlds, i.e. estimating low or even very low probabilities, which can be problematic in practice. The classical approach (Philip et al., 2020; Naveau et al., 2020) is to make use of results from Extreme Value Theory (EVT) in order to estimate a parametric probability distribution based on past observations or outputs from climate models. EVT shows (Coles et al., 2001) that the distribution of block maxima — typically yearly maxima for climate data — of a random variable converges towards

a universal distribution called the Generalized Extreme Value (GEV) distribution which has three parameters: the location $\mu$, scale $\sigma$ and shape $\xi$ parameters. The existence of this mathematical result suggests to fit a GEV distribution on yearly maxima of $X$, taking into account the non-stationarity of the climate system by letting, for example, the location parameter $\mu$ depend on a measure of global warming such as Global Mean Surface Temperature (GMST) or Regional Mean Surface Temperature (RMST) (Naveau et al., 2020; Robin and Ribes, 2020): $\mu(\text{RMST}) = \mu_0 + \mu_1 \text{RMST}$. It is then possible to compare

the probabilities of reaching the level $x$ of the observed event with RMST in the counterfactual and RMST in the factual world and to compute the associated probability ratio. The expected *intensity change* of the event between the two worlds can also be estimated as $\Delta X = \mu_1 \times (\text{RMST}(F = 1) - \text{RMST}(F = 0))$, although this expression for the intensity change has to be adapted when other hypotheses are made on the GEV parameters to take into account the non-stationarity of the climate system.

This method is *unconditional* in the sense that it is purely statistical and gives absolute probabilities for the yearly maxima

of the observable $X$ of interest. Whatever the actual dynamics of the observed event, it compares its intensity with the yearly maximum intensities of the past. It therefore bears the risk of comparing events that were yearly maxima but that had different dynamical mechanisms. To alleviate this issue, several *conditional* methods have been proposed (Yiou et al., 2017; Terray, 2021; de Vries et al., 2024; Leach et al., 2024). These methods have in common to condition the attribution analysis on the large scale synoptic pattern $\mathcal{C}$ associated to the observed event. As a consequence, they address the question of the mean

changes between the counterfactual and factual worlds for events dynamically similar to the one observed: $\Delta_{\mathcal{C}} X = \mathbb{E}[X \mid F = 1, \mathcal{C}] - \mathbb{E}[X \mid F = 0, \mathcal{C}]$ where each expectation is conditional on the large scale synoptic pattern $\mathcal{C}$ of the event. In this sense, these methods condition on the dynamics to isolate the thermodynamical signal. Conditional methods are also useful to explore the physical causes of changes in the extremes, one of the key elements to support the results of attribution studies.

This *conditional* attribution framework allows to answer the question: how a similar large-scale circulation pattern in the

two worlds leads to different outcomes in an observable of interest? If the difference $\Delta_{\mathcal{C}} X$ is statistically significant, then one can say that in the factual world the event has been rendered more (or less) intense by $\Delta_{\mathcal{C}} X$. The *unconditional* probability



ratio and intensity change can in principle be obtained (Yiou et al., 2017) if one can estimate the probabilities $\mathbb{P}[\mathcal{C} \mid F]$ and $\mathbb{P}[\mathcal{C} \mid X \geq x]$ in the two worlds. Estimating these two probabilities is however very difficult in practice because of the under-sampling of synoptic scale patterns similar to $\mathcal{C}$ in limited size data sets. Moreover, the *conditional* probability ratio is also not
provided by these methods.

    Here we use the conditional flow analogues attribution methodology proposed by Faranda et al. (2022) and adapted from Yiou et al. (2017). This method is for example used by the ClimaMeter tool developed to provide rapid attribution results (Faranda et al., 2023a). We propose a procedure to compute both conditional intensity changes and probability ratios when using analogues of the synoptic circulation. We illustrate the procedure on three recent impactful events in Europe: the 25th of
July 2019 heatwave event in North-Western Europe, the 11th of February 2020 wind event in Ireland and the UK and the 4th of October 2021 precipitation event in the Italian Alps. The synoptic situations and the observables considered for the three events are presented in Figure 1. The 25th of July 2019 event was characterized by a strongly meridional meander of the mid-level jet which leads to exceptional hot extremes in Northern France, Belgium, Western Germany and Southern England (Fig. 1a, see also Vautard et al. (2020)). The 11th of February 2020 event coincided with the present of storm Ciara in Western of Europe
and lead to important wind damages in Ireland, the UK, France, Belgium and the Netherlands (Fig. 1b, Galvin (2022)). Finally, the 4th of October 2021 was an extreme Meditterranean episode which lead to intense precipitations in the North of Italy and South-East of France (Fig. 1c, Cassola et al. (2023)).

    The paper is organized as follows. In section 2 we present the data used and the method employed. We especially detail the hypotheses of the method and the statistical procedure employed to test the significance of the results obtained. The results are
presented in section 3. We discuss these results and the limits of the method in section 4. Finally, the conclusions are drawn in section 5.

## 2   Data and methods

### 2.1   Data

For all the analyses presented here we use the ERA5 reanalysis data set over the period 1950-2021 (Hersbach et al., 2020). We
consider daily mean fields for the geopotential height at 500hPa ($z_{500}$), 2m air temperature ($t2m$), 10m wind speed ($wind_{10}$) and we use a 5-day rolling mean for total daily precipitations ($tp$). Note that the ERA5 reanalysis procedure does not assimilate precipitation data and can present important biases with respect to observational data sets (Lavers et al., 2022; Xu et al., 2022). The absolute values provided here must therefore be taken with care. We nevertheless choose to use ERA5 precipitation data for consistency with the other fields and because this paper proposes a methodological development rather than a formal attribution
study. We regrid the original 0.25° ERA5 resolution to a 1° resolution for the fields studied here. The reason for using such a lower resolution is that, with the analogue method and using a limited size data set, the analogues found can be slightly shifted horizontally. This would also correspond to an horizontal shift of the observables of interest and therefore we cannot expect to reconstruct properly their distributions at the original 0.25° resolution.





For estimating trends, we regress quantities of interest on the Regional Mean Surface Temperature (RMST) rather than the
Global Mean surface Temperature (GMST). We make this choice to encompass the local warming trend which can result from
additional mechanisms compared to the GMST, for example aerosols concentrations and land-cover changes (Robin and Ribes,
2020; Schumacher et al., 2024). RMST is computed as the area weighted average of $t2m$ between 35°N-70°N and 15°W-30°E
over both land and ocean and we then apply twice a 11-years rolling mean as a low pass filter to obtain a smoothed time series.
We note that this simple procedure tends to underestimate the actual warming of Europe for the last years of the time series
because there is no data for the years after 2023 in our data set to compute the rolling mean. This would therefore make our
results conservative when concluding on the anthropogenic influence on the events studied.

## 2.2 Analogues attribution: computation of intensity change and probability ratio

The flow analogues attribution methodology is based on the idea of finding in the past synoptic circulation patterns — called
*analogues* — similar to the one observed for the extreme event and comparing the hazards they produce. Here we look for
analogues of the synoptic circulation using geopotential height at 500hPa ($z_{500}$) for the three events as it acts as an approximate
streamfunction of the free troposphere atmospheric circulation. There is a positive trend in the geopotential height field which
reflects the warming of the atmosphere and can lead to finding inappropriate analogues. To avoid this effect, we detrend
uniformly the geopotential height field against RMST. We emphasize that doing so only shifts vertically the $z_{500}$ patterns
while keeping the correct latitudinal and longitudinal gradients from which the winds can be derived using the geostrophic
approximation.

The analogues are found over the period 1950-2021 — without separating in two periods as in Faranda et al. (2022) —
and using the domains shown in Figure 1 (dashed boxes): 30°N-68°N and 20°W-25°E for the temperature event, 35°N-65°N
and 25°W-20°E for the wind event, 30°N-65°N and 20°W-25°E for the precipitation event. These domains are chosen using
our own expert judgment to find the synoptic structures associated with the events considered, as is customary in attribution
studies. We explore the sensitivity of the results obtained in the following by shrinking and expanding these domains by 3°
of latitude and longitude at each edge. For our analysis, we take the 72 best analogues as the synoptic patterns minimizing
the pointwise Euclidean distance with respect to the synoptic pattern of the event. This is equivalent to finding approximately
one analogue per year, although we emphasize that we do not impose that one analogue per year has to be found (there can
be several analogues per year). We only impose that the analogues should be separated by at least 5 days and we exclude the
event itself as an analogue. In order to take into account the seasonal cycle and to be close to the observed event, we impose
that the analogues must be found in certain months: June to August for the temperature event, October to March for the wind
event, September to November for the precipitation event. We also test the sensitivity of the results to the number of analogues
found by increasing and decreasing the number of analogues by 25% (54 and 90 analogues).

To check the quality of the analogues found with our procedure, as in Faranda et al. (2022), we compute the analogues
quality metric $Q$. For each event and for each analogue $k$ of each event, its quality $Q^k$ is computed as the average Euclidean
distance of its own $n = 72$ analogues computed over the same domain and time period. Note that when we determine the
analogues of each analogue $k$ of the event, they need not a priori be the same as the ones of the event itself. In other words,



the fact that a pattern $A$ is among the $n = 72$ best analogues of a pattern $B$ does not always imply that the pattern $B$ is among

the $n = 72$ best analogues of the pattern $A$. We then compare the value of the analogues quality $Q^{\text{event}}$ for the event with the

125 distribution of the $(Q^k)_{1 \leq k \leq n}$ of its analogues. If $Q^{\text{event}}$ is a clear outlier of the distribution of $(Q^k)_{1 \leq k \leq n}$ — for example if

it is higher than their maximum — this means that the synoptic pattern of the event is unique and therefore that it has bad

analogues. In this case, a conditional (and probably also unconditional) attribution statement is likely impossible based on past

data only.

As a result of this procedure, we have for each event $n = 72$ analogues dates and therefore $n$ observables of interest

(temperature, wind, precipitations) $(X_{i,j}^k)_{1 \leq k \leq n}$ at each grid point $i, j$. For these $n$ dates, we also have $n$ values of RMST:

$(\text{RMST}^k)_{1 \leq k \leq n}$. As in the EVT-based approach, the results obtained will crucially depend on the hypothesis made to relate

the $(X_{i,j}^k)_{1 \leq k \leq n}$ and the $(\text{RMST}^k)_{1 \leq k \leq n}$. Here, we follow the practice of current attribution methods (Philip et al., 2020) and

we assume a linear link for temperature and wind:

$$X_{i,j}^k = \alpha_{i,j} + \beta_{i,j} \text{RMST}^k + \epsilon_{i,j} \tag{1}$$

and a log-linear link for precipitations:

$$\ln X_{i,j}^k = \alpha_{i,j} + \beta_{i,j} \text{RMST}^k + \epsilon_{i,j}, \tag{2}$$

the latter expressing a Clausius-Clapeyron-like relationship between global/regional temperatures and precipitations. The $\epsilon_{i,j}$

are random terms on which we make our parametric assumptions (see below). The $\alpha_{i,j}$ and $\beta_{i,j}$ are then determined by

Ordinary Least-Square (OLS) regression. The *intensity change* $\text{IC}_{i,j}$ at grid point $i, j$ is therefore computed as:

$$\text{IC}_{i,j} = \beta_{i,j} \times (\text{RMST}_{\text{event}} - \text{RMST}_{1950}) \tag{3}$$

for temperature and wind, and as:

$$\text{IC}_{i,j} = X_{i,j}^{\text{event}} - X_{i,j}^{\text{event}} e^{\beta_{i,j}(\text{RMST}_{1950} - \text{RMST}_{event})} \tag{4}$$

for precipitations. Here $\text{RMST}_{\text{event}}$ is the RMST for the event (assumed to be the factual world) and $\text{RMST}_{1950}$ is the RMST

in 1950 (assumed to be the counterfactual world). For precipitations, $X_{i,j}^{\text{event}}$ is the intensity of the event at grid point $i, j$.

At this stage, we detrend the time series $(X_{i,j}^k)_{1 \leq k \leq n}$ with the coefficients determined above to obtain a new time series

$(\tilde{X}_{i,j}^k)_{1 \leq k \leq n}$. This times series is an empirical sampling (up to the detrending procedure) of the distribution $\tilde{X}_{i,j} \mid \mathcal{C}$ of the

observable $\tilde{X}_{i,j}$ conditional on the synoptic situation $\mathcal{C}$ of the extreme event. If we had enough data — i.e. a longer data set and

therefore more analogues — we could give an empirical estimation of the probability to reach level $\tilde{X}^{\text{event}}$ for this conditional

distribution. Considering that $\tilde{X}^{\text{event}}$ is likely extreme — even when conditioning on the synoptic pattern $\mathcal{C}$ — it is usually

not possible to give a precise empirical value to this probability. We therefore need to use a parametric hypothesis to estimate

probabilities in both worlds. The difficulty is that, contrary to the EVT-based method, we have no a priori choice for what

distribution $\tilde{X}_{i,j} \mid \mathcal{C}$ should follow. Here, we propose to use the Skew-Normal distribution for temperature and wind and the

Gamma distribution for precipitations (see sections 3 and 4 for a discussion on the choice of these probability distributions).





We fit a Skew-Normal or Gamma distribution on the $(\tilde{X}_{i,j}^k)_{1 \leq k \leq n}$ using the method of moments. The choice of this method for

the fits was made to speed up computations considering the large number of fits necessary with our bootstrap procedure (see below and appendix A for the detail of the method of moments for the Skew-Normal and Gamma distributions). Once we have access to the parameters of the distribution, we evaluate the probability of $X_{i,j}^{\text{event}}$ for these distributions in the counterfactual world and in the factual world. For the Skew-Normal distribution, this procedure means shifting the fitted location parameter by $\beta_{i,j}\text{RMST}_{1950}$ (counter-factual world) and $\beta_{i,j}\text{RMST}_{\text{event}}$ (factual world). For the Gamma distribution, this is equivalent to

multiplying the scale parameter by $e^{\beta_{i,j}\text{RMST}_{1950}}$ (counter-factual world) and $e^{\beta_{i,j}\text{RMST}_{\text{event}}}$ (factual world).

The full procedure can be summarized as follows:

1. Find $n$ analogues over the period 1950-2021,

2. Extract the $n$ values of the observable $X$ at grid point $i, j$: $(X_{i,j}^k)_{1 \leq k \leq n}$,

3. Extract the $n$ values of RMST: $(\text{RMST}^k)_{1 \leq k \leq n}$,

4. Detrend the time series $(X_{i,j}^k)_{1 \leq k \leq n}$ with respect to $(\text{RMST}^k)_{1 \leq k \leq n}$: linearly for temperature and wind, log-linearly for precipitations,

5. Compute the intensity change as the difference between the event and the projection of the event in 1950 based on the detrending hypothesis,

6. Fit a parametric distribution on the detrended time series $(\tilde{X}_{i,j}^k)_{1 \leq k \leq n}$: a Skew-Normal distribution for temperature and

wind, a Gamma distribution for precipitations,

7. Compute the probabilities for the present of the event and 1950 using the fitted distribution and evaluate the probability ratio $\mathbb{P}(X \geq x \mid F = 1)/\mathbb{P}(X \geq x \mid F = 0)$.

We assess the sensitivity of the results obtained using a bootstrap procedure: at each grid point $i, j$ we resample $10^3$ times $n$ values of the analogues observables $(X_{i,j}^k)_{1 \leq k \leq n}$ with replacement and execute the procedure described above. We report

the median value of the $10^3$ resamplings. This result is said to be significant if the critical value (0 for the intensity change and 1 for the probability ratio) is either above the 97.5th quantile or below the 2.5th quantile of the $10^3$ resamplings. The same procedure is applied to each grid point $i, j$, all grid points being treated independently.

## 2.3   EVT-based attribution

We additionally compare our procedure to the unconditional EVT approach. To do so, at each grid point $i, j$ we compute

the yearly maxima of the observable of interest, restricting over the same months as the ones over which the analogues are computed (June to August for the temperature event, October to March for the wind event, September to November for the precipitation event). We then detrend this time series of yearly maxima and compute the intensity change similarly as presented above for the analogues procedure. We emphasize however that there is no a priori reason that the intensity change computed on





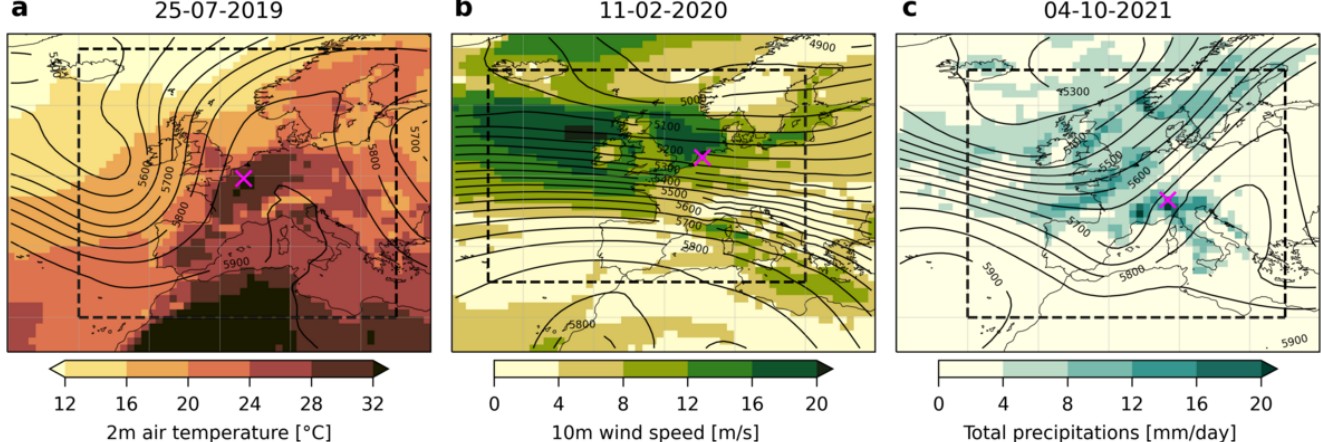

**Figure 1.** Synoptic situation and observable considered for the three events studied. (a) Geopotential height at 500hPa (m, contours) and 2-m air temperature (°C, colours) for the 25th of July 2019, (b) geopotential height at 500hPa (m, contours) and 10-m wind speed (m/s, colors) for the 11th of February 2020 and (c) geopotential height at 500hPa (m, contours) and total precipitations averaged over 5 days (mm/day, colors) for the 4th of October 2021. For all events the black dashed box shows the region where the analogues are computed and the magenta cross shows the grid point taken as example in Figure 3.

the yearly maxima should be similar to the one for the analogues distribution because the yearly maxima may not correspond to

185 analogues of the event. We then fit a GEV distribution on the detrended time series using the method of L-moments (Hosking, 1990). The probabilities in the counterfactual and factual worlds are recovered as above by shifting the location parameter by $\beta_{i,j}\mathrm{RMST}_{1950}$ and $\beta_{i,j}\mathrm{RMST}_{event}$ for temperature and wind, and multiplying the location and scale parameters by $e^{\beta_{i,j}\mathrm{RMST}_{1950}}$ and $e^{\beta_{i,j}\mathrm{RMST}_{event}}$ for precipitations. From these probabilities we recover the probability ratio. To estimate the sensitivity of these results, we also use a bootstrap procedure with $10^3$ resamplings and we report the median result when it is significant.

We note that this procedure is not exactly the same as the one of the World Weather Attribution (Philip et al., 2020), which fits directly a non-stationary GEV with the maximum likelihood method. We made this choice here in order to be more similar to our procedure for the analogues based attribution. Nonetheless, this procedure can be adapted straightforwardly if one wants to use the maximum likelihood method with non-stationary Skew-Normal, Gamma and GEV distributions.

# 3 Results

## 195 3.1 Illustration with three grid points

To investigate the relevance of the analogues found for the three events, we show in Figure 2abc the distributions of the quality of the analogues found (boxplots) and the analogues quality for the event itself (red dot). All events are in the upper tail of the distribution of analogues quality of their analogues — which may be expected in-so-far as they are all rare events — but are





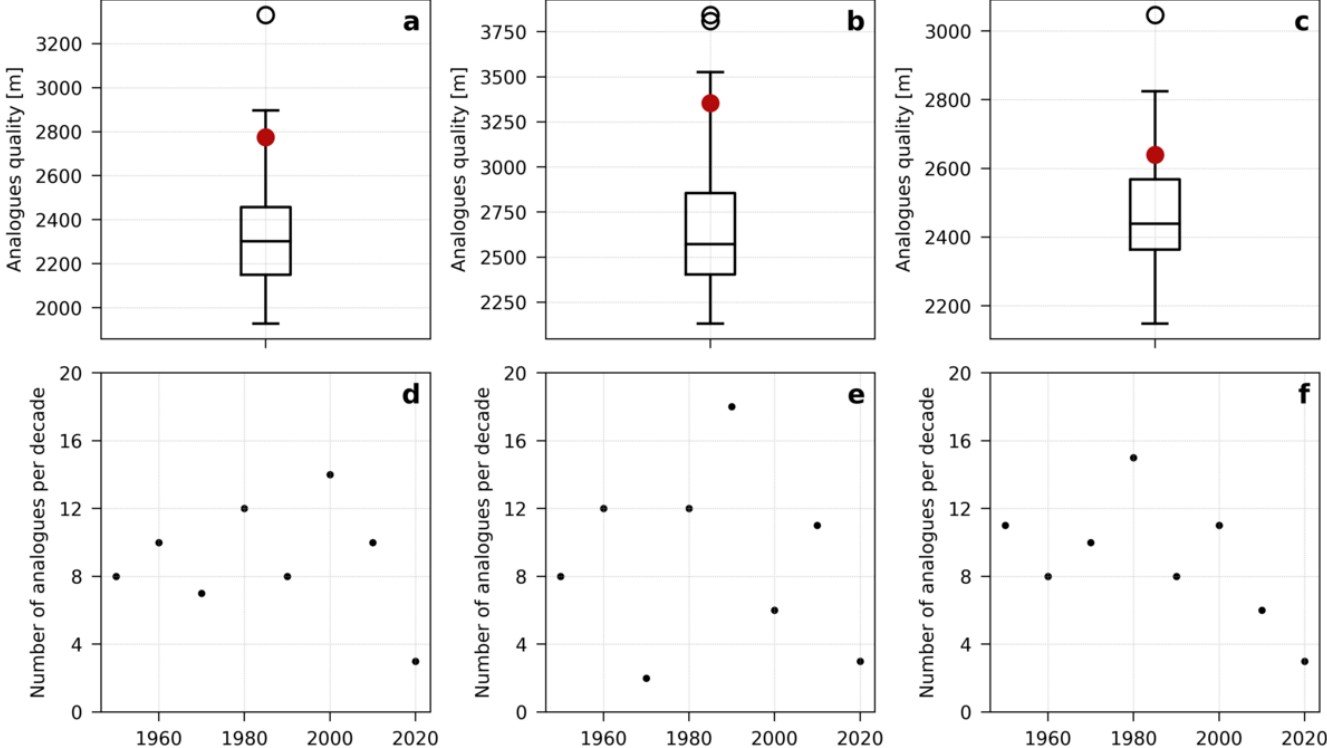

**Figure 2.** Analogues quality and trend in the number of analogues per decade. First row: distribution of the analogues quality over the 72 analogues for (a) the 25th of July 2019 event, (b) the 11th of February 2020 event and (c) the 4th of October 2021 event. The boxplots show the 25th and 75th quantiles and the median of the distribution. For each plot the red dot shows the analogues quality for the event itself. Second row: number of analogues per decade for (d) the 25th of July 2019 event, (e) the 11th of February 2020 event and (f) the 4th of October 2021 event. For every panel, we show the trend in the number of analogues per decade if the trend is statistically significant at the 5% level using a bootstrap procedure (see text for the detail).

not outliers of the distributions. There are 4 analogues with worst analogues quality for the temperature event, 3 for the wind event and 12 for the precipitation event. Figure 2def show the number of analogues per decade for each event. We compute the linear trend over this number of analogues to explore whether they have become more likely with *time*. The significance of the trend is computed using a bootstrap procedure: we resample $10^3$ times 72 analogues and compute the trend. If the values 0 is outside the 95% interval centered around the median of these trends, then the trend is said to be significant. Note that because we have only 2 years in the 2020 decade, these years are not taken into account for computing the trend. For none of the events this trend is significant which shows that the analogues are well distributed over the period 1950-2021.

In order to give probability ratios measuring the increasing/decreasing likelihood of the extreme events considered, we need to rely on a parametric hypothesis for the distribution of the observable conditional on the synoptic pattern of the event. If we had a much larger sample size than the ERA5 reanalysis provides, we could compute empirical probabilities and this could for





example be done with either a long run or a large ensemble of a climate model. Contrary to the EVT approach to attribution,
which is based on a theoretical result providing which parametric distributions should be used for computing probability ratios,
here we have no a priori theoretical basis for which distribution to use. Figure B1 shows the empirical skewness (first row)
and excess kurtosis (second row) for the analogues distributions of the observables for the three events considered (after the
detrending procedure). We assess the statistical significance of these quantities using the same bootstrap procedure as the one
described for the analogues attribution (see section 2) and show in white grid points which are not statistically significant.
These results should nevertheless be taken with care as the precise estimation of the third and fourth moment with $n = 72$ is
difficult. For temperature and wind (Fig. B1 panels a, b, d and e), most of the grid points do not show a significant departure
from 0, i.e. from the third and fourth moments of a Gaussian distribution. Temperature tends to be negatively skewed over sea
surfaces (panel a) and wind tends to be negatively skewed over sea surfaces in the North of Europe and positively skewed over
the Mediterranean land surfaces (panel b), but for both observables, the departure from 0 is small over the regions of interest for
both events, except over the North Sea for the wind event. Similarly, the excess kurtosis is not different from 0, and if anything
it tends to be slightly negative – although this may arise as the result of under-sampling the very extremes. As a consequence,
we propose to use the Skew-Normal distribution — which is a modification of the Gaussian distribution to take into account
the skewness of the distribution (see appendix A) — to represent the analogues distribution of these observables at each grid
point.

For precipitation (Fig. B1 panels c and f) the results are different. The conditional distribution is significantly positively
skewed for most grid points, as expected for an observable which is bounded downwards by 0, and the excess kurtosis is
not significantly different from 0 for most land grid points. For grid points where the excess kurtosis is significant, it is
strongly positive which would be likely to be the same for most grid points if we had more analogues in so far as precipitation
distributions are usually long tailed. As a consequence, we treat differently the precipitation observable and we use a Gamma
distribution to fit its distribution. The Gamma distribution is commonly used to fit precipitations data (Stagge et al., 2015;
Gudmundsson and Seneviratne, 2016; Martinez-Villalobos and Neelin, 2019). We come back to the question of the choice of
these distributions in the discussion section.

Figure 3 shows an illustration of our method on three example grid points marked by a magenta cross in Figure 1. For the
25th of July 2019 temperature event, the intensity reached is so extreme that it is never reached by its analogues, even after
detrending the analogues observables of the past (panel a). As a consequence, it is in the far tail of the fitted distributions for
both the past (1950) and the present (2019, panel d) and the probability ratios are largely higher than 1, with a median value
around $10^5$. Accordingly, the median intensity change is around 4.5°C. For the 11th of February 2020 wind event, the event
itself was intense but four analogues in the past show higher values than the event. The trend of the analogues observables with
respect to RMST is weak, and as a consequence, the conditional distributions in the past and the present are close. The median
probability ratio is around 0.7 and the median intensity change around -0.5 m/s, but none of them are statistically significant
according to the bootstrap procedure. For the 4th of October 2021 precipitation event, the event itself was also intense but
exceeded by seven analogues in the past. The logarithmic regression points towards a decrease in the intensity of this events,
which largely decreases the intensity of the analogues events as projected in 2021 (panel c). The event therefore becomes very





unlikely in the present (panel f) and the median probability ratio is around 0.01 with an intensity change around -30 mm/day,
both being statistically significant. To test the parametric hypothesis made for computing the probability ratios we employ the
two-sided Kolmogorov-Smirnov test on each resampled time series to test the hypothesis that the distribution of the resampled
time series is different from the fitted distribution, i.e. a Skew-Normal distribution for temperature and wind and a Gamma
distribution for precipitation. Using the 5% confidence level, 99.8% resampled time series are not distinguishable from the
target distribution for the temperature event, 100% for the wind event and 98% for the precipitation event. This shows that the
chosen distributions are compatible with the data.

## 3.2 Results in Europe

We apply the same procedure to every grid points in Europe. The results for the median probability ratios and intensity changes
are shown in Figure 4. For the intensity changes, in addition to the observable considered, we also show the significant changes
in the synoptic field (geopotential height at 500hPa). The temperature event is associated to significant increases of probability
ratios over Western Europe, especially in the North of France, Belgium, the Netherlands and the South of England where they
are higher than $10^3$ (panel a). The intensity changes show a similar pattern (panel d), although with interesting differences:
whereas the intensity changes are similar in the North-East of Spain to the one in the North of France, Belgium and the
South of England, the probability ratios of the former are smaller. This likely reflects different scales and shapes of the fitted
distributions of the observables. The wind event presents no changes in the probability ratios. In the main zone of interest for
the event (Ireland, the UK and the North Sea), no grid points show a significant change (panel b) which is also reflected in the
intensity changes (panel e). For the precipitation event, for the region of interest (Italian Alps), grid points where the extremes
occurred show a probability ratio significantly below 1, from 0.1 to 0.01 (panel c). The corresponding intensity changes are
accordingly negative from -8 to -20 mm/day. Finally, we note that the changes of the synoptic fields for all events are minimal
or nonexistent: there is a small increase in the intensity of the anticyclone over the ocean west of Brittany for the temperature
event, no changes for the wind event and a small decrease of the geopotential height over the ocean west of Morocco for the
precipitation event. Figure B2 in appendix shows at each grid point the proportion of resampled time series which pass the
Kolmogorov-Smirnov test at the 5% level, i.e. the proportion of the resampled time series for which the proposed distribution
is a correct representation of the empirical distribution. For the temperature and events (Fig. B2ab), over the regions of interest
this proportion is higher than 90% and close to 100% for most grid points. For the precipitation event (Fig. B2c) the proportion
is also close to 100% in the Italian Alps, except for one specific grid point.

To test the sensitivity of these results, we present in appendix similar figures to Figure 4 where we change the number of
analogues by ±25% (54 and 90 analogues over 1950-2021, Figures B3 and B4) and the size of the domain to find analogues by
±3° of longitude and latitudes at the edge of the domains defined in Figure 1 (Figures B5 and B6). For the temperature event,
the results presented above are stable to both a change in the number of analogues and the size of the domain (except maybe
when we use 90 analogues). This likely reflects the strong warming in extreme temperatures observed in Western Europe and
already discussed by several previous works (Vautard et al., 2023; Patterson, 2023; Noyelle et al., 2023). For the wind event,
the results obtained tend to be similar: there is no detectable changes, except maybe a small decrease in the intensity of winds





**Figure 3.** Illustration of the analogues attribution method for the three example grid points of Figure 1. First row (a,b,c): raw values of the analogues observables (black dots) and detrended values of the analogues observables (red dots) vs RMST. The observables are detrended to correspond to the RMST of the event. The black plain line and the black shadings show the median regression line and the 95% uncertainties interval for the regression of the raw values of the analogues observables against RMST. The horizontal black dashed line shows the intensity of the event. Note that for panel c the regression is logarithmic with respect to RMST (see section 2). Second row: fitted (d and e) Skew-Normal and (f) Gamma distributions for RMST in 1950 and in the present (i.e. for the event). The plain line shows the median fit and the shadings the 95% uncertainty interval obtained after bootstrap (see section 2). The empirical histogram corresponds to the detrended values of the analogues observables. The vertical black dashed line shows the intensity of the event. Third row (g,h,i): bootstrap distribution of probability ratios and intensity changes for the events. For the probability ratios, the black horizontal line shows the value 1 (no probability change). For the intensity changes, the black horizontal line shows the value 0 (no intensity change).





**Figure 4.** Probability ratios and intensity changes with the analogues method. First row (a,b,c): median probability ratios obtained at each grid point for (a) the temperature event, (b) the wind event and (c) the precipitation event. The grid points are colored in white when the the probability ratios is not statistically different than 1 (see section 2). Second row (d,e,f): median intensity changes for the observable (colors) and the synoptic field ($z_{500}$ in m, contours). For the observable, the grid points are colored in white when the the intensity change is not statistically different than 0 (see section 2). For the synoptic field, the intensity change is not shown when it is not statistically different than 0.

for some grid points. Finally, for the precipitation event in the Italian Alps, there is still a general decrease in the intensity of precipitations when changing both the number of analogues and the domain size but the significance of this decrease can disappear for some grid points when changing these parameters.





### 3.3 Comparison with the climatological and EVT-based approaches

The intensity changes presented in Figure 4 are computed conditional on the analogues, i.e. on the synoptic pattern of the extreme events observed. To investigate the difference with respect to conditioning on the analogues compared to other methods, we present in Figure 5abc the intensity changes deduced from the climatological trends computed on the months when the analogues are found. At each grid point, we compute the intensity change as previously but this time the trend is computed by considering all days of the months when the analogues are searched (June to August for the temperature event, October to March for the wind event, September to November for the precipitation event). Figure 5def show the difference with the analogues intensity changes when they are statistically significant according to the bootstrap procedure as previously. For the temperature and precipitations events, the intensity changes conditional on the analogues over the regions of interest for the extremes are stronger than the climatological trends, and even the reverse for the precipitation events. This demonstrates the contribution of conditioning on the analogues. For the wind events there are weak or no trends in the region of interest for both the climatology and the analogues. We could in principle do the same analysis for probability ratios but we would need to make a parametric assumption on the full distribution of the observable over the months considered, which is likely more difficult than for the distribution conditional on the analogues.

Finally, we compare our results with the classical EVT-based approach for extreme events. The EVT approach is unconditional and compares the intensity of the event observed to a non-stationary GEV distribution fitted on the yearly maxima over the period 1950-2021. Figure 6 shows the results with this approach for the three grid points studied in Figure 3. For the temperature event, both approaches conclude to an increasing probability and intensity of this event but the EVT approach provides lower probability ratios and intensity changes (Fig. 3g vs Fig. 6g). For the wind event, both approaches give similar, non significant results for the grid point selected (Fig. 3h vs Fig. 6h). Lastly, for the precipitation event the EVT approach gives a non significant result but would point towards an increase in the probability and intensity of this event, contrary to the analogues approach (Fig. 3i vs Fig. 6i).

Figure 7 shows the equivalent of Figure 4 with the EVT approach. It displays the median probability ratios and intensity changes found with the EVT approach after $10^3$ resamplings when they are statistically significant at each grid point. The results obtained are rather different from the analogues approach. For the temperature event, the significant probability ratios tend to be confined to Western Europe (especially France) and the intensity changes are smaller than the ones observed with the analogues method. For the wind event, only few and sparse grid points show a significant change. The precipitation event also does not show significant changes over most of Europe and especially in the Italian Alps, contrary to the analogues method. Note that for precipitations, it is not clear that yearly maxima have converged towards a GEV distribution and it may be more suitable to use larger block sizes (Alaya et al., 2020), although this would reduce the sample sizes for the fits.

### 4 Discussion

In this paper we propose to estimate intensity changes and probability ratios for the flow analogues extreme events attribution method. The main improvement compared to the method proposed by Faranda et al. (2022) is to avoid the arbitrary split of the





**Figure 5.** Climatological intensity changes and comparison with analogues intensity changes. First row (a,b,c): median climatological intensity changes based on the trend over the months where the analogues are searched for. Second row (d,e,f): difference between climatological intensity changes and analogues intensity changes of Figure 4. The grid points are colored in white when the difference is not statistically significant according to the bootstrap procedure (see section 2).

analogues in two periods. Doing so increases the number of analogues found and therefore gives more statistical strength to
the results obtained, even though we have to make some additional statistical assumptions.

Our procedure estimates intensity changes by regressing the observables of interest on a metric measuring anthropogenic
global warming — Regional Mean Surface Temperature (RMST) here. We applied this method grid point by grid point, but it
could be applied for example over a spatial average to study a particular region of interest. The hypotheses made to estimate
intensity changes are minimal and unrelated to the parametric assumption for the computation of the probability ratios. The
results obtained can thus be considered as a good approximation of the response to increasing RMST of the mean of the
observables of interest conditional on the synoptic pattern of the event of interest (as soon as there are stable to small changes







**Figure 6.** Same as Figure 3 with the EVT approach.

in the number of analogues and the domain to compute analogues). The intensity changes thus give an estimate of the mean observed thermodynamical response for a particular synoptic scale pattern.

The hypotheses made to estimate probability ratios on the other hand are more problematic because they rely on a parametric
approximation. If the fitted distributions for the conditional distribution of the observable are incorrect, this could lead to large errors in the estimated probability ratios, especially when the extreme event studied is largely outside the distribution of its analogues (such as for the 25th of July 2019 temperature event). Here we presented arguments based on the third and fourth moments of the conditional distributions to justify the use of Skew-Normal distributions for temperature and wind and Gamma distributions for precipitations. We nevertheless acknowledge that these arguments are based on empirical results and have to
be tested case by case. As a consequence, it is likely that the choice of the fitted distributions can be questioned and could



**Figure 7.** Same as Figure 4 with the EVT approach.

be adapted to find more suitable distributions for the estimation of probability ratios for other events and other observables. Moreover, even with our parametric choice, as illustrated here in Figure 3 and Figure 6 the range of uncertainties on the probability ratios can be several orders of magnitude large according to the bootstrap procedure. This is a known problem for risk-based extreme events attribution method which arises as a result of the estimation of a ratio of low or very low probabilities. It is however not even clear that the mean value of this ratio is well defined statistically — for example if the mean probability of the event in the counterfactual world is equal to 0. For these reasons, it is probably more meaningful and cautious to use intensity changes rather than probability ratios for reporting attribution results. As a consequence, the parametric hypotheses made to represent the probabilities conditional on the synoptic pattern may not be that important to establish an attribution statement for extreme events, as soon as the intensity change is significantly different from 0.



Similar to the EVT-based attribution method, the flow analogues method may suffer from an under-sampling of extremes due
to the use of a limited size reanalysis data set only. This method can straightforwardly be used with climate models outputs to
strengthen the analysis, especially with large ensembles to find more analogues. Conditioning on a measure of global warming
as done here could also allow to compare results for models with different climate sensitivities. However, models have known
deficiencies, including biases (Maraun, 2016; Vrac et al., 2023; François et al., 2020) and incorrect dynamics of extremes
under forcing over Western Europe (**?**Patterson, 2023; Vautard et al., 2023; d'Andrea et al., 2024) which may not alleviate the
sampling issue of the reanalysis. Another sampling issue concerns the natural variability of the climate system. If a long term
physical phenomenon has covaried over a long period of time with RMST and can influence the intensity of the events studied,
then this would lead to an incorrect estimation of the impact of global warming per se. Using analogues over 72 years, as done
here, partially alleviates this risk compared to separating in two periods as in Faranda et al. (2022), especially when they are
well distributed over time with little increasing or decreasing trends (Fig. 2). Another way to circumvent this issue could be
to include measures of natural variability on the regressions of the observable, for example the Atlantic Meridional Variability
(AMV) for extremes in Europe (Suarez-Gutierrez et al., 2023).

We nevertheless want to note that the main drawbacks of the method presented here are also common to the classical EVT
method, namely: under-sampling, representation of natural variability, use of past observations vs model outputs. The inter-
pretation of the results of the two approaches are also different. The EVT-based approach gives the probability that the yearly
maximum of an observable is above a given level and therefore the probability ratio gives how this probability has changed
between the factual and counterfactual worlds. It thus encompasses both the dynamical changes — increasing frequency of
certain weather patterns caused by anthropogenic global warming (Vautard et al., 2023; Faranda et al., 2023b; Dong et al.,
2024; d'Andrea et al., 2024) — and the thermodynamical changes for the strongest extremes. As a consequence, its analysis
may be far from the actual extreme event observed and in particular can always make an attribution statement even though
the dynamics of the event has never been observed in the past at the place considered (Faranda et al., 2023a). The flow ana-
logues method on the other hand gives the change in the probability of a certain level given the synoptic pattern, as soon as the
synoptic pattern has good analogues in the past. This method separates the dynamical contribution from the thermodynamical
contribution. It does not address the unconditional probability of reaching an extreme — which may be the most interesting
aspect for the general public — but, as shown here, it tends to give a better attribution signal because thermodynamical changes
are likely more easily detectable than dynamical changes (Shepherd, 2014; Vautard et al., 2023). Our results suggest that the
EVT-based approach may tend to be too conservative in its attribution statements by considering only the strongest extremes
for which rare or very rare dynamical mechanisms may overrun the climate change thermodynamical signal. As illustrated
here, the two methods do not give the same absolute results in general, and may also give opposing results as shown for the
precipitation event. It is not clear though which method we should prefer for an attribution statement, especially when they
provide opposing conclusions.



## 5 Conclusions

In this paper we proposed a way to compute intensity changes and probability ratios for the flow analogues extreme events methodology proposed by Faranda et al. (2022) and adapted from Yiou et al. (2017). Contrary to Faranda et al. (2022), we do not separate the data sets in two periods but we search for analogues of the synoptic pattern of the extreme event on the full data set (1950-2021). We then fit a linear model on the analogues observable of interest to estimate intensity changes with increasing Regional Mean Surface Temperature (RMST). We compute probability ratios by making a parametric hypothesis on the distribution of the observables conditional on the synoptic scale pattern. We finally estimate the sensitivity of the results by a bootstrap procedure and report the median values when they are statistically significant. The method proposed here can be applied to other observables of interest and using outputs of climate models. One advantage of the method proposed here is that we condition on a measure of global warming, which, when applied with model outputs, would allow to compare models with different climate sensitivities.

We illustrate the method on three recent events in Europe: the 25th of July 2019 temperature event, the 11th of February 2020 wind event and the 4th of October 2021 event. We find that the intensity changes for the temperature event over Western Europe are around 4.5°C and the probability ratios above $10^3$. These results are stable to a change in the parameters of the method which makes possible to say that this event was made more likely and more intense under climate change. The intensity changes and probability ratios over Ireland, the UK and the North Sea for the wind event do not detect any change and this result seems robust to specifications, which suggests that this event was not impacted by AGW. Lastly, the precipitations event in the Italian Alps and South-East of France tends to be less likely and intense under climate change, but the results are also sensitive to the specification of the method. For the wind and precipitation events, our results with the analogues method are at odds with the results obtained with the EVT-based method. All these attribution statements are to be understood as conditional to the synoptic scale pattern observed during the events.

*Code and data availability.* ERA5 reanalysis data are available on the Copernicus website (https://cds.climate.copernicus.eu). The code to obtain the results presented here is available at the following link [to be added after revision]



## Appendix A: Method of moments for the Skew-Normal and Gamma distributions

### A1  Skew-Normal distribution

The Skew-Normal distribution is an adaptation of the Gaussian distribution to account for non-zero skewness. Its PDF can be expressed as:

$$f(x) = \phi(\frac{x-\mu}{\sigma})\Phi(\xi\frac{x-\mu}{\sigma}) \tag{A1}$$

where $\phi(x) = \frac{1}{\sqrt{2\pi}}e^{-\frac{1}{2}x^2}$ is the PDF of the standard normal, $\Phi(x) = \int_{-\infty}^{x}\phi(t)dt$ is the CDF of the standard normal and $\mu$, $\sigma$ and $\xi$ are the location, scale and shape parameters of the distribution.

The method of moments take a simple analytic form for this distribution. Let us note $\hat{\gamma}_1$ the empirical skewness (centered and normalized third order moment) of the samples $(X^k)_{1\leq k\leq n}$. We define:

$$\hat{\delta} := \sqrt{\frac{\pi}{2}\frac{|\hat{\gamma}_1|^{2/3}}{|\hat{\gamma}_1|^{2/3} + (\frac{4-\pi}{2})^{2/3}}}. \tag{A2}$$

From this value, we can derive an estimation of the three parameters of the distribution:

$$\begin{cases} \hat{\xi} = \dfrac{\hat{\delta}}{\sqrt{1-\hat{\delta}^2}} \\[2ex] \hat{\sigma} = \dfrac{\hat{s}}{\sqrt{1-2\hat{\delta}^2/\pi}} \\[2ex] \hat{\mu} = \hat{m} - \hat{\sigma}\hat{\delta}\sqrt{\dfrac{2}{\pi}} \end{cases} \tag{A3}$$

where $\hat{m}$ and $\hat{s}$ are the empirical mean and standard deviation of the samples $(X^k)_{1\leq k\leq n}$. Note that for the Skew-Normal distribution, the skewness has a maximum absolute value close to 0.99. When applying the method of moments here we therefore take:

$$|\hat{\gamma}_1| = \min\left(0.99, \sum_{k=1}^{n}(\frac{X^k-\hat{m}}{\hat{s}})^3\right) \tag{A4}$$

and $\hat{\delta}$ has the same sign as $\hat{\gamma}_1$.

### A2  Gamma distribution

Here we use the Gamma distribution defined on $[0, +\infty[$, i.e. with a null location parameter. The PDF of the distribution is:

$$f(x) = \frac{1}{\Gamma(\xi)\sigma^\xi}x^{\xi-1}e^{-x/\sigma} \tag{A5}$$

where $\Gamma$ is the Gamma function, $\xi$ is the shape parameter and $\sigma$ the scale parameter.





The method of moments also take a simple analytic form for this distribution:

$$
\begin{cases}
\hat{\sigma} = \dfrac{\hat{s}^2}{\hat{m}} \\[2mm]
\hat{\xi} = \hat{m}/\hat{\sigma}
\end{cases}
\tag{A6}
$$

where $\hat{m}$ and $\hat{s}$ are the empirical mean and standard deviation of the samples $(X^k)_{1 \leq k \leq n}$.







**Figure B1.** Empirical skewness (first row) and excess kurtosis (second row) of the analogues distribution of observables for the three events considered. Only the grid points where the skewness and excess kurtosis are significantly different than 0 are shown (see text for the detail).

**Appendix B: Supplementary figures**



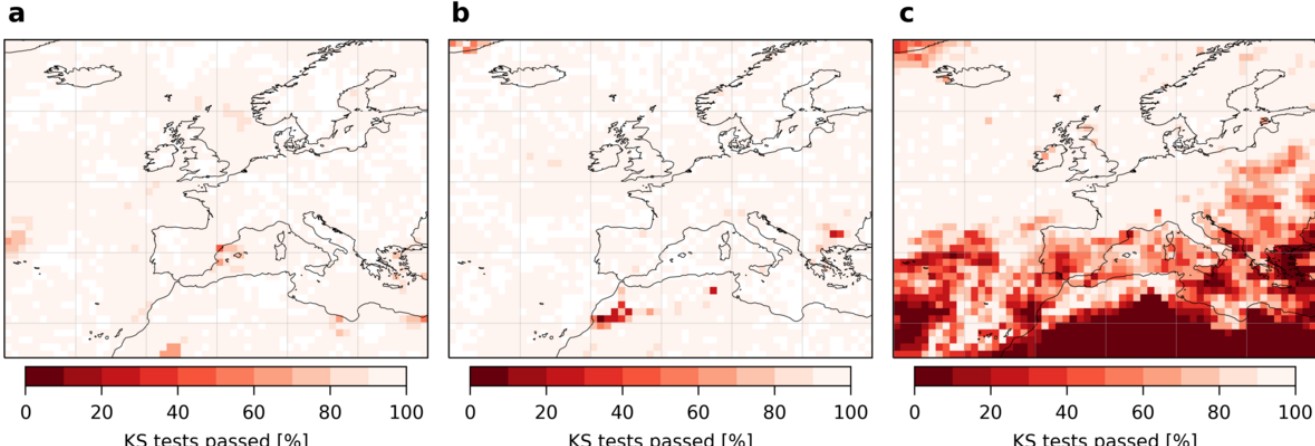

**Figure B2.** Kolmogorov-Smirnov tests. Proportion of Kolmogorov-Smirnov tests passed at the 5% level over the $10^3$ resamplings at each grid point of the analogues distribution of (a) the 25th of July 2019 temperature event, (b) the 11th of February 2020 wind event and (c) the 4th of October 2021 precipitation event.

*Author contributions.* RN proposed the method and did the data analysis. All authors advised the data analysis and participated to the writing of the paper.

*Competing interests.* The authors declare no conflict of interest.

*Acknowledgements.* This paper received support from the grant ANR-20-CE01-0008-01 (SAMPRACE), and from the European Union's Horizon 2020 research and innovation programme under grant agreement No. 101003469 (XAIDA), from the European Union's Horizon
2020 Marie Sklodowska-Curie grant agreement No. 956396 (EDIPI), from the LEFE-MANU-INSU-CNRS grant "CROIRE", from the "COESION" project funded by the French National program LEFE (Les Enveloppes Fluides et l'Environnement) and from state aid managed by the National Research Agency under France 2030 bearing the references ANR-22-EXTR-0005 (TRACCS-PC4-EXTENDING project).

We acknowledge useful discussions within the MedCyclones COST Action (CA19109) and the FutureMed COST Action (CA22162) communities.





**Figure B3.** Same as Figure 4 with 54 analogues.

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





**Figure B4.** Same as Figure 4 with 90 analogues.

d'Andrea, F., Duvel, J.-P., Rivière, G., Vautard, R., Cassou, C., Cattiaux, J., Coumou, D., Faranda, D., Happé, T., Jézéquel, A., et al.: Summer deep depressions increase over the Eastern North Atlantic, Geophysical Research Letters, 51, e2023GL104 435, 2024.

de Vries, H., Lenderink, G., van Meijgaard, E., van Ulft, B., and de Rooy, W.: Western Europe's extreme July 2019 heatwave in a warmer world, Environmental Research: Climate, 2024.

Dong, C., Noyelle, R., Messori, G., Gualandi, A., Fery, L., Yiou, P., Vrac, M., D'Andrea, F., Camargo, S. J., Coppola, E., et al.: Indo-Pacific
regional extremes aggravated by changes in tropical weather patterns, Nature Geoscience, pp. 1–8, 2024.

Faranda, D., Bourdin, S., Ginesta, M., Krouma, M., Messori, G., Noyelle, R., Pons, F., and Yiou, P.: A climate-change attribution retrospective of some impactful weather extremes of 2021, 2022.

Faranda, D., Messori, G., Coppola, E., Alberti, T., Vrac, M., Pons, F., Yiou, P., Saint-Lu, M., Hisi, A., Brockmann, P., et al.: ClimaMeter: Contextualising Extreme Weather in a Changing Climate, Weather and Climate Dynamics, 2023a.



**Figure B5.** Same as Figure 4 with an analogues domain larger by 3° of latitude and longitude at each edge.

Faranda, D., Messori, G., Jezequel, A., Vrac, M., and Yiou, P.: Atmospheric circulation compounds anthropogenic warming and impacts of climate extremes in Europe, Proceedings of the National Academy of Sciences, 120, e2214525 120, 2023b.

François, B., Vrac, M., Cannon, A. J., Robin, Y., and Allard, D.: Multivariate bias corrections of climate simulations: which benefits for which losses?, Earth System Dynamics, 11, 537–562, 2020.

Galvin, J.: The storms of February 2020 in the channel islands and south west England, Weather, 77, 43–48, 2022.

Gudmundsson, L. and Seneviratne, S. I.: Anthropogenic climate change affects meteorological drought risk in Europe, Environmental Research Letters, 11, 044 005, 2016.

Hersbach, H., Bell, B., Berrisford, P., Hirahara, S., Horányi, A., Muñoz-Sabater, J., Nicolas, J., Peubey, C., Radu, R., Schepers, D., et al.: The ERA5 global reanalysis, Quarterly Journal of the Royal Meteorological Society, 146, 1999–2049, 2020.

Hosking, J. R.: L-moments: analysis and estimation of distributions using linear combinations of order statistics, Journal of the Royal

Statistical Society Series B: Statistical Methodology, 52, 105–124, 1990.





**Figure B6.** Same as Figure 4 with an analogues domain larger by 3° of latitude and longitude at each edge.

Lavers, D. A., Simmons, A., Vamborg, F., and Rodwell, M. J.: An evaluation of ERA5 precipitation for climate monitoring, Quarterly Journal of the Royal Meteorological Society, 148, 3152–3165, 2022.

Leach, N. J., Roberts, C. D., Aengenheyster, M., Heathcote, D., Mitchell, D. M., Thompson, V., Palmer, T., Weisheimer, A., and Allen, M. R.: Heatwave attribution based on reliable operational weather forecasts, Nature Communications, 15, 4530, 2024.

Maraun, D.: Bias correcting climate change simulations-a critical review, Current Climate Change Reports, 2, 211–220, 2016.

Martinez-Villalobos, C. and Neelin, J. D.: Why do precipitation intensities tend to follow gamma distributions?, Journal of the Atmospheric Sciences, 76, 3611–3631, 2019.

National Academies of Sciences Engineering and Medicine: Attribution of Extreme Weather Events in the Context of Climate Change, The National Academies Press, Washington, DC, https://doi.org/10.17226/21852, 2016.

Naveau, P., Hannart, A., and Ribes, A.: Statistical methods for extreme event attribution in climate science, Annual Review of Statistics and Its Application, 7, 89–110, 2020.



Noyelle, R., Zhang, Y., Yiou, P., and Faranda, D.: Maximal reachable temperatures for Western Europe in current climate, Environmental Research Letters, 18, 094 061, 2023.

Patterson, M.: North-West Europe Hottest Days Are Warming Twice as Fast as Mean Summer Days, Geophysical Research Letters, 50, e2023GL102 757, 2023.

Philip, S., Kew, S., van Oldenborgh, G., Otto, F., Vautard, R., van der Wiel, K., King, A., Lott, F., Arrighi, J., Singh, R., et al.: A protocol for probabilistic extreme event attribution analyses. Adv Stat Climatol Meteorol Oceanogr 6: 177–203, 2020.

Robin, Y. and Ribes, A.: Nonstationary extreme value analysis for event attribution combining climate models and observations, Advances in Statistical Climatology, Meteorology and Oceanography, 6, 205–221, 2020.

Schumacher, D. L., Singh, J., Hauser, M., Fischer, E. M., Wild, M., and Seneviratne, S. I.: Exacerbated summer European warming not captured by climate models neglecting long-term aerosol changes, Communications Earth & Environment, 5, 182, 2024.

Seneviratne, S. I., Zhang, X., Adnan, M., Badi, W., Dereczynski, C., Di Luca, A., Ghosh, S., Iskander, I., Kossin, J., Lewis, S., et al.: Weather and climate extreme events in a changing climate (Chapter 11), 2021.

Shepherd, T. G.: Atmospheric circulation as a source of uncertainty in climate change projections, Nature Geoscience, 7, 703–708, 2014.

Stagge, J. H., Tallaksen, L. M., Gudmundsson, L., Van Loon, A. F., and Stahl, K.: Candidate distributions for climatological drought indices (SPI and SPEI), International Journal of Climatology, 35, 4027–4040, 2015.

Stott, P. A., Christidis, N., Otto, F. E., Sun, Y., Vanderlinden, J.-P., van Oldenborgh, G. J., Vautard, R., von Storch, H., Walton, P., Yiou, P., et al.: Attribution of extreme weather and climate-related events, Wiley Interdisciplinary Reviews: Climate Change, 7, 23–41, 2016.

Suarez-Gutierrez, L., Müller, W. A., and Marotzke, J.: Extreme heat and drought typical of an end-of-century climate could occur over Europe soon and repeatedly, Communications Earth & Environment, 4, 415, 2023.

Terray, L.: A dynamical adjustment perspective on extreme event attribution, Weather and Climate Dynamics, 2, 971–989, 2021.

Van Loon, S. and Thompson, D. W.: Comparing Local Versus Hemispheric Perspectives of Extreme Heat Events, Geophysical Research Letters, 50, e2023GL105 246, 2023.

Vautard, R., van Aalst, M., Boucher, O., Drouin, A., Haustein, K., Kreienkamp, F., Van Oldenborgh, G. J., Otto, F. E., Ribes, A., Robin, Y., et al.: Human contribution to the record-breaking June and July 2019 heatwaves in Western Europe, Environmental Research Letters, 15, 094 077, 2020.

Vautard, R., Cattiaux, J., Happé, T., Singh, J., Bonnet, R., Cassou, C., Coumou, D., D'Andrea, F., Faranda, D., Fischer, E., et al.: Heat extremes in Western Europe are increasing faster than simulated due to missed atmospheric circulation trends, 2023.

Vrac, M., Thao, S., and Yiou, P.: Changes in temperature–precipitation correlations over Europe: are climate models reliable?, Climate Dynamics, 60, 2713–2733, 2023.

Xu, J., Ma, Z., Yan, S., and Peng, J.: Do ERA5 and ERA5-land precipitation estimates outperform satellite-based precipitation products? A comprehensive comparison between state-of-the-art model-based and satellite-based precipitation products over mainland China, Journal of Hydrology, 605, 127 353, 2022.

Yiou, P., Jézéquel, A., Naveau, P., Otto, F. E., Vautard, R., and Vrac, M.: A statistical framework for conditional<? xmltex \newline?> extreme event attribution, Advances in Statistical Climatology, Meteorology and Oceanography, 3, 17–31, 2017.