# Peer review of "Attributing the occurrence and intensity of extreme events with the flow analogues method"

_EGUsphere, 2024_

## Referee Comment (RC1)

Review of 'Attributing the occurrence and intensity of extreme events with the flow analogues method'

In this paper, the authors tackle the problem of attributing extreme events to climate change. They take the angle of conditioning extreme events on the regional circulation, as captured by flow analogues, over Europe in the considered cases. This has the advantage of allowing a more 'apples-to-apples' comparison of dynamically similar events, in contrast to the more standard GEV approach based on annual block maxima.
The paper is methodologically focused, and aims to demonstrate the validity and usefulness of this flow-dependent approach on three high profile European extreme events from recent years, and compares the findings to both an alternate detrending procedure and a version of the GEV/ETV approach. They also explore the sensitivity of their method to parameter variations. I think the approach is very interesting; it is definitely sensible and insightful to integrate circulation information into the attribution process.

However the conclusions the authors' analysis provides for the given cases are rather unusual, and after a few careful reads I think the authors either need to better evidence the plausibility of their findings or revise their method. Therefore I am suggesting major revisions. I detail my objection immediately below, followed by a number of minor and typographical comments.

To quickly summarise what the analogue-based attribution shows for each case study:
- The heatwave was made ~4 degrees warmer by climate change, and ~100,000x more likely. The EVT approach claims a similar intensity change (>3 degrees), but only deems the event 10-100x more likely.
- No significant change in probability ratio or intensity of the wind extreme. The EVT approach seems to indicate reduced probability/intensity, but not significantly.
- The precipitation event was made ~100x less likely and was made 8-20 mm/day weaker (or 20-50mm/day weaker; there is an inconsistency between text and figure 3i). The EVT approach suggests a slight, not very significant, increase in probability and intensity.

The precipitation case gives me the most cause for concern. Looking at how the logarithmic fit in 3c,f squeezes the distirbution so drastically (transforming a >25mm/day event into a <10mm/day event in the current climate), it seems very hard to believe this is a sensible transformation to apply. Under the EVT approach in fig 6f meanwhile, the change in distribution is small. To me this makes sense given the fairly weak apparent correlation between RMST and total precip in 3c.
Coming from another perspective, and thinking about the meteorology, the 4 Oct 2021 event was a result of an extratropical cyclone tracking across the Mediterranean, with the anticyclonic anomaly to the east helping to stall its eastward movement. Do we really expect the probability of extreme rainfall when a cyclone hits the coast to have decreased by 90-99% relative to 1950? Or indeed, to have had its intensity ~halved? This runs counter to basic theory (Clausius-Clapeyron relation) and previous work that find little observational change in cyclone properties.
It seems to me that either the logarithmic method of trend fitting is producing spurious results, and/or the similarity metric used to define analogues is not faithful to the driving meteorology (e.g. the analogues don't include cyclones when the true flow does). I suspect that defining analogues based solely on Z500 over a large area is too coarse an approach.

I am also a bit worried about the high probability ratios estimated for the heat case. You mention later that probability ratios have many issues, but the fact remains that your method produces PR estimates 1000x that of the EVT approach. I don't think this is unrelated from the fact that the EVT distribution is positive skewed. While you've fitted a skew normal for the analogues, the distribution has ended up very gaussian so the estimation of tail probabalities is going to be very

unreliable. Again, there could be a possible issue with the analogues: it wouldn't take a large distortion of the flow field to put central france under a trough.

I ask the authors to revise or more fully justify these more implausibel findings I've mentioned. It might aid interpretation to actually plot ~4 analogues (say the 15$^{th}$, 30$^{th}$, 45$^{th}$ and 60$^{th}$ closest analogues) for each case in the appendix, to see if a mismatch of meteorology between the event and analogues does indeed explain this.

Minor/typographical comments
- Line 12: what is a 'climatological event'? A little unclear to me. Also 'negatively affect' reads more naturally than 'affect negatively'
- Line 15: 'which aim  to'
- Line 31: Given 'very low' doesn't have a well defined quantitative meaning, I'd just say 'estimating low probabilities'
- Line 48: 'These methods both condition the ...'
- Line 57: I think by this point in the introduction it would be good to explicitly state the assumption of your work that changes in the synoptic dynamics themselves due to climate change are negligible/ should be treated separately. The conditional attribution is informative but by definition only a partial attribution to thermodynamic changes. As far as I can see this is not written in the introduction, and should be clearly emphasised.
- Line 62: 'This method is used, for example, by the...'
- Line 68: 'Which lead to exceptional heat in...'
- Line 69: 'Western  Europe'
- Line 71: 'precipitation' (and several other times in the document)
- Line 81: Can you motivate why you take a 5 day mean for precip? I presume to focus on the synoptic driving.
- Line 86: Even 1 degree is a fairly strong constraint on shifts. It might be interesting to see results for a  2.5/3 deg box average centred on the three points examined in figure 3.
- Line 103: Clearer to say "thickening due to warming of the atmosphere".
- Line 111: Its not totally clear in what space you compute the analogues. The full high-dimensinal space of gridpoint values within each region? That's giving you a O(1000) dimensional space. Given that distance vectors in high-dimensional space tend towards equal length (https://link.springer.com/chapter/10.1007/3-540-44503-X_27) which may lead to poor ability to detect analogues, did you consider computing analogues in an O(10-100) dimensional PCA space?
- Line 127: impossible from the thermodynamic perspective, but this could be tackled by considering how synoptic conditions are being impacted by climate change.
- Line 136: What is the motivation for using surface temperature for something like precip, given condensation happens aloft? If RMST is used just for simplicity then maybe make a comment about this somewhere.
- Line 199: Worst-> worse
- Fig 3abc: You should state in the figure caption that the 'detrended' data is shifted to the 2020 RMST value. It makes sense for your application, but isn't a standard detrending, where you'd expect to end up with zero mean displacement of the data.
- Line 263: The intensity change your report does not correspond to what you show in fig 3i.
- The captions for figures B5 and B6 are the same.
- Line 345: missing reference.

---

## Author Comment (AC1)

**Reviewer 2**

This paper presents a methodology for attributing extreme weather events using flow analogues, using three different event types within Europe (precipitation, heat, and wind) as worked examples. The method is compared to the 'traditional' probabilistic attribution method. The paper provides arguments for the methodological choices in the analogues attribution technique, stressing both the advantages compared to GEV and advantages compared to earlier uses of analogues. The approach methodology presented is well, and is a great development for the use of analogues in attribution.

However, I have a few major comments that I think need addressing, followed by some more minor comments.

**We thank the reviewer for their comments. In the following we address them.**

1. I think more consideration of the results and how/why they initially appear to (in some cases) contradict the GEV method needs to be made. More thought and discussion on the 'attribution statement', and importantly the framing of the questions that each attribution methods can be used for, needs to be made. A related point which is not discussed in detail in the paper, is the advantage of the method presented over other conditional methods (e.g. those referenced in line 48), and how using a range of methods to provide multiple lines of evidence could be useful. As is, the paper appears to pit the analogues method against GEV, rather than consider how it is useful to consider both to increase understanding.

This could be a useful reference - Coumou, D., Arias, P.A., Bastos, A., Gonzales, C.K.G., Hegerl, G.C., Hope, P., Jack, C., Otto, F., Saeed, F., Serdeczny, O. and Shepherd, T.G., 2024. How can event attribution science underpin financial decisions on Loss and Damage?. *PNAS nexus*, *3*(8), p.pgae277.

**We thank the reviewer for suggesting this interesting reference which is indeed very relevant for our discussion. We have included at multiple places in the paper comments to answer the questions raised by the reviewer:**
**a. Framing of the questions:**
**i. Introduction L43-60: "This method is unconditional in the sense that it is purely statistical and gives absolute probabilities for the yearly maxima of the observable X of interest. Whatever the actual dynamics of the observed event, it compares its intensity with the yearly maximum intensities of the past. It therefore bears the risk of comparing events that were yearly maxima but that had different dynamical mechanisms. To alleviate this issue, several conditional methods have been proposed [...] they address the question of the mean changes between the counterfactual and factual worlds for events dynamically similar to the one observed:[...] In this sense, these methods condition on the dynamics to isolate the thermodynamical signal. Conditional methods are also useful to explore the physical causes of changes in the extremes, one of the key elements to support the results of attribution studies.**

**This conditional attribution framework allows to answer the question: how a similar large-scale circulation pattern in the two worlds leads to different outcomes in an observable of interest? [...]"**

ii. Discussion L 359-374: "We nevertheless want to note that the main drawbacks of the method presented here are also common to the classical EVT method, namely: under-sampling, representation of natural variability, use of past observations vs model outputs. The interpretation of the results of the two approaches are also different. The EVT-based approach gives the probability that the yearly maximum of an observable is above a given level and therefore the probability ratio gives how this probability has changed between the factual and counterfactual worlds. It thus encompasses both the dynamical changes --- increasing frequency of certain weather patterns caused by anthropogenic global warming \citep{vautard2023heat, faranda2023atmospheric, dong2024indo, d2024summer} --- and the thermodynamical changes for the strongest extremes. As a consequence, its analysis may be far from the actual extreme event observed and in particular can always make an attribution statement even though the dynamics of the event has never been observed in the past at the place considered \citep{faranda2023climameter}. The flow analogues method on the other hand gives the change in the probability of a certain level given the synoptic pattern, as soon as the synoptic pattern has good analogues in the past. This method separates the dynamical contribution from the thermodynamical contribution. It does not address the unconditional probability of reaching an extreme --- which may be the most interesting aspect for the general public --- but it tends to give a better attribution signal because thermodynamical changes are likely more easily detectable than dynamical changes \citep{shepherd2014atmospheric, vautard2023heat}. Our results suggest that the EVT-based approach may tend to be too conservative in its attribution statements by considering only the strongest extremes for which rare or very rare dynamical mechanisms may overrun the climate change thermodynamical signal."

b. Advantage of our method: L388: "However, conditional attribution methods, such as the one presented here or others \citep{yiou2017statistical, terray2021dynamical, de2024western, leach2024heatwave}, are more focused on the very dynamics of the event observed and may provide a more detectable (thermodynamical) signal. The main advantage of our method is to use only past data and therefore to avoid common pitfalls of modeling studies."

c. Multiple lines of evidence: L390: "As argued recently by \citet{coumou2024can}, using a range of methods to provide multiple lines of evidence for an attribution statement useful for practitioners is therefore absolutely necessary."

2. For the precipitation event many of the analogues show little rainfall (particularly after detrending). This suggests that the analogues poorly represent the precipitation of the observed event. Can you show that the analogues do represent the precipitation adequately? (This could also be needed for temperature and wind, but it appears less of an issue for these variables – perhaps suggesting the method is only suitable for specified event types).

**We thank the reviewer for this suggestion and we have now included it in the paper (Figure 1). We have also changed the region where the analogues of the precipitation event were looked for as the resulting precipitation field did not reflect the event itself.**

**We also refer the reviewer to our answer to reviewer 1 for the particular case of the precipitation event on the points raised.**

3. One point that is mentioned in the discussion (line 363), but I feel should be stressed further, is that the method only works when there are good analogues. Further discussion of this would be valuable – are there certain event types better suited to the method? Or some events (perhaps hurricanes) where the event should not be used due to insufficient past analogues (though maybe large model ensemble could be used instead)? What if good analogues only occur in later decades?

**The reviewer raises interesting points that may need further investigation. We have discussed partly some of the points raised, but without a more systematic investigation we prefer not to give general conclusions which may reveal not to be true. We feel like the quality metric we provided and the discussion on the quality of the composite fields (L197-203) for the analogues provides evidence for how to determine whether good analogues are found for a particular case study. To answer more precisely the questions raised by the reviewer:**
      **a. Are certain event types better suited to the method? -> it is likely that synoptic scale events with minor contributions from mesoscale structures – such as heatwaves for example — are likely more suited for the method because better analogues are likely to be found.**
      **b. Insufficient past data and use of large model ensemble -> we indeed agree with the reviewer and have discussed these points in L347-352**
      **c. Good analogues in later decades -> if such a thing appears then it would point towards an increasing probability of the synoptic pattern of the event. This should not impact the conditional probability statement but would qualitatively suggest an increasing unconditional probability of the event. However, validating such a condition quantitatively is likely difficult (as mentioned in L57-58) and subject to the possibility that the increasing recurrence of this synoptic pattern only arises as a result of natural variability.**

**Minor comments**

1. The 'past' starts in 1950, restricted by the ERA5 dataset. Do you think this matters? Do you think results would differ significantly if you were able to go right back to 1850?

**It would definitely be more interesting to start the analysis at an earlier date, simply because it would imply having more data points and therefore improve the quality of the statistical analysis. However, as can be seen in Figure 3abc, the analysis lacks data points for high RMST, i.e. for the present with global warming. It is therefore probably more this period that would benefit from more sampling. Finally, going back to 1850 would allow us to sample more the natural variability of the climate system but, given that we already use 70 years of data, we do not expect to have large gains from going this far in the past. As a consequence, we do not expect that going much farther in the past would significantly change the results obtained.**

2. Line 65 – dates formatting, could remove the 'of''s throughout (e.g. 4th July, not 4th of July)

**This has been corrected.**

3. Line 98 – why do you take Z500 rather than SLP (as used in Faranda et al 2024)?

**The choice of using Z500 in this study is justified by previous studies (e.g. Faranda et al. 2020 or Jezequel et al. 2018). We also note that Faranda et al. 2024 use Z500 and SLP alternatively. Using Z500 implies conditioning on the mid-tropospheric circulation while using SLP implies conditioning on the surface circulation. In general, one or the other choice can be made depending on expert knowledge of what is the most relevant for the extreme event studied (or even both choices can be examined to compare the results). Here we chose not to explore the sensitivity of our results to this choice.**

4. Line 112 - Some other analogue studies use spatial correlation to identify analogues, why did you choose to use Euclidean distance?

**In general, our experience with the use of analogues is that the distance used to find analogues is not a major determinant of the analogues found (Yiou, 2014). As a consequence, we could indeed have used spatial correlation (note that maximizing spatial correlation is the same as minimizing Euclidean distance if the data is spatially normalized). We do not expect to have major differences in the results obtained if we had used spatial correlation to find the analogues (we tested the method using the L1 norm instead of the L2 norm and found no major differences).**

5. Line 200 – Assessing linear trend per decade – there aren't many data points, did you test sensitivity shifting the decades (i.e. 1955-1964, 1965-1974 or other starting years)?

**We agree with the reviewer that there are not many data points and therefore that the assessment of the trend should be taken with some caution. To test its sensitivity we employed a bootstrap procedure on the analogues rather than shifting the decade as suggested by the reviewer (L209-212). We thank the reviewer for this alternative method but as it is not the core of the results obtained in the paper we prefer to keep the method we proposed.**

6. Fig.2 caption – the final sentence is a bit misleading as no trends are shown. I think this should be removed, and just referred to in the results (I spent a while trying to spot the trend!)

**This has been removed as suggested by the reviewer.**

7. Fig 2 / 3 / 6 – it would be great to title the columns as you do in Fig4

**This has been modified as suggested by the reviewer.**

8. Fig3/6 could you align the zeros?

**We are not sure to understand which zeros the reviewer is referring to. In both panels abc and panels ghi, the range of values and units used are different for each plot, therefore, for ease of visualization we prefer to keep the figure as it is (note that in panels ghi the zeros are indicated by an horizontal line).**

9. Lin 255 - 'northern France and 'southern England' rather than north of etc.

**This has been corrected.**

10. Line 271 – You chose to use the same number of analogues for all events, but some events may have more good analogues (i.e. be less dynamically extreme). Would there be a way to incorporate this into the method, so use a different number of analogues depending how good the analogues are? Maybe by using the information in Fig2a,b,c?

**The reviewer raises an interesting point. The choice of the number of analogues we made is indeed somewhat arbitrary (although we tested the sensitivity of our results to this choice). It would be very relevant to have an absolute criteria to define what is a good or bad analogue. Unfortunately we are not aware of a method to do such a procedure systematically. As a consequence, the best we can recommend is probably to proceed as we did by systematically evaluating the sensitivity of the results to the number of analogues chosen. We prefer also to Platzer et al. (2021) for recommendations on the number of analogues for simple models.**

11. Fig.4 - z500 contours are not very clear to me

**This is indeed the case because for most of them the difference is not significant. Making them more clear to read would imply modifying the results displayed, that is why we had to display them as they are. Although we agree that this is an issue for visualization we prefer to keep them as such with the associated commentary in the text.**

12. Line 279 – precipitations (no need for 's')

**This has been corrected.**

13. Line 345 – stray '?' in references

**This has been corrected.**

14. Line 392 (final sentence) This doesn't quite read right to me, consider rewording.

**We have modified the sentence as follows: L412 " All these attribution statements are conditional to the synoptic scale pattern observed during the events."**

**References**

Jézéquel, A., Yiou, P., & Radanovics, S. (2018). Role of circulation in European heatwaves using flow analogues. *Climate dynamics*, *50*(3), 1145-1159.

Yiou, P. (2014). AnaWEGE: a weather generator based on analogues of atmospheric circulation. *Geoscientific Model Development*, *7*(2), 531-543.

Faranda, D., Vrac, M., Yiou, P., Jézéquel, A., & Thao, S. (2020). Changes in future synoptic circulation patterns: consequences for extreme event attribution. *Geophysical Research Letters*, *47*(15), e2020GL088002.

Platzer, P., Yiou, P., Naveau, P., Filipot, J. F., Thiébaut, M., & Tandeo, P. (2021). Probability distributions for analog-to-target distances. *Journal of the Atmospheric Sciences*, *78*(10), 3317-3335.

---

## Author Comment (AC2)

**Reviewer 1**

In this paper, the authors tackle the problem of attributing extreme events to climate change. They take the angle of conditioning extreme events on the regional circulation, as captured by flow analogues, over Europe in the considered cases. This has the advantage of allowing a more 'apples-to-apples' comparison of dynamically similar events, in contrast to the more standard GEV approach based on annual block maxima.

The paper is methodologically focused, and aims to demonstrate the validity and usefulness of this flow-dependent approach on three high profile European extreme events from recent years, and compares the findings to both an alternate detrending procedure and a version of the GEV/ETV approach. They also explore the sensitivity of their method to parameter variations. I think the approach is very interesting; it is definitely sensible and insightful to integrate circulation information into the attribution process.

However the conclusions the authors' analysis provides for the given cases are rather unusual, and after a few careful reads I think the authors either need to better evidence the plausibility of their findings or revise their method. Therefore I am suggesting major revisions. I detail my objection immediately below, followed by a number of minor and typographical comments.

**We thank the reviewer for considering our work positively. We share part of the concerns expressed by the reviewer with regards to the results displayed. In the following we address their comments.**

To quickly summarise what the analogue-based attribution shows for each case study:
- The heatwave was made ~4 degrees warmer by climate change, and ~100,000x more likely. The EVT approach claims a similar intensity change (>3 degrees), but only deems the event 10-100x more likely.
- No significant change in probability ratio or intensity of the wind extreme. The EVT approach seems to indicate reduced probability/intensity, but not significantly.
- The precipitation event was made ~100x less likely and was made 8-20 mm/day weaker (or 20-50mm/day weaker; there is an inconsistency between text and figure 3i). The EVT approach suggests a slight, not very significant, increase in probability and intensity. The precipitation case gives me the most cause for concern. Looking at how the logarithmic fit in 3c,f squeezes the distribution so drastically (transforming a >25mm/day event into a <10mm/day event in the current climate), it seems very hard to believe this is a sensible transformation to apply.

Under the EVT approach in fig 6f meanwhile, the change in distribution is small. To me this makes sense given the fairly weak apparent correlation between RMST and total precip in 3c.

Coming from another perspective, and thinking about the meteorology, the 4 Oct 2021 event was a result of an extratropical cyclone tracking across the Mediterranean, with the anticyclonic anomaly to the east helping to stall its eastward movement. Do we really expect the probability of extreme rainfall when a cyclone hits the coast to have decreased by 90-99% relative to 1950? Or indeed, to have had its intensity ~halved? This runs counter to basic theory (Clausius-Clapeyron relation) and previous work that find little observational

change in cyclone properties. It seems to me that either the logarithmic method of trend fitting is producing spurious results, and/or the similarity metric used to define analogues is not faithful to the driving meteorology (e.g. the analogues don't include cyclones when the true flow does). I suspect that defining analogues based solely on Z500 over a large area is too coarse an approach.

**We agree with the reviewer that the results obtained for the precipitation event are implausible with respect to the current knowledge of the change in the dynamics of cyclones and high precipitations in the Mediterranean. We investigated the two points raised by the reviewer:**

**(i) after inspection of the analogues values for precipitation for several grid points, we saw no evidence that the logarithmic method of trend fitting would be incompatible with the structure of the data (not shown)**

**(ii) however, when looking at the synoptic composite maps of both the Z500 and precipitation for the analogues found with our initial choices, it was clear that the structure of the precipitation event was not recovered by the analogues (see Figure 1 showing this structure for the new analogues).**

**As suggested by the reviewer, this is indeed likely because the area chosen for finding analogues for this event was too large. In the updated version of the manuscript we reduced this area to be closer to the region where the event actually occurred. This led to a much reduced estimation of the negative intensity changes (see Figure 3 and Figure 4), which is more coherent with the results obtained with the GEV. Despite the fact that the analogues quality metric for this event is good compared to the two other events (Figure 2), the synoptic field is not as similar as the event itself compared to the two other events (Figure 1f). We therefore agree that the results obtained for this event should be taken with care - similarly to those obtained with the GEV method though. We discuss those points more extensively in the result and discussion section (see below).**

**One may wonder why the results obtained here do not seem to follow the expected increase of precipitation with increasing temperature following the Clausius-Clapeyron law. To explore this discrepancy, we show below (Figure R1) the intensity change of the 2m air temperature obtained using the analogues of the precipitation event. In most of the area concerned with the high precipitations (south of the Alps), there is no significant change in the temperature field. This is coherent with no significant change in the precipitation field for these analogues. Why the analogues of this event are not associated with an increasing temperature is not clear though. We checked whether the analogues in the more recent years are found in a climatologically colder part of the year (shift in the seasonality of the analogues), but found no evidence for such a shift (not shown). It suggests that the warming signal depends on the atmospheric circulation, which also highlights the interest of conditioning on large scale circulation. These elements are now mentioned in the text: L379-382.**

[Figure]

**Figure R1: Intensity change of the 2m air temperature for the analogues of the precipitation event**

I am also a bit worried about the high probability ratios estimated for the heat case. You mention later that probability ratios have many issues, but the fact remains that your method produces PR estimates 1000x that of the EVT approach. I don't think this is unrelated from the fact that the EVT distribution is positive skewed. While you've fitted a skew normal for the analogues, the distribution has ended up very gaussian so the estimation of tail probabilities is going to be very unreliable. Again, there could be a possible issue with the analogues: it wouldn't take a large distortion of the flow field to put central France under a trough.

**We agree with the reviewer that the large difference in the PR in our approach compared to the EVT approach is due to the positively skewed temperature distribution fitted using a GEV. We first note that this difference in the skewness reflects the difference in the skewness of the original data, which is largely due to one data point only: the event itself in 2019. If we remove this data point from the fit, the**

skewness is not as pronounced and the median PR goes from around 5 to around 200, denoting the sensitivity to just one data point of tail probabilities estimated using a GEV distribution (see Figure R2). Therefore, the estimation of these tail probabilities both with our method and the GEV method is problematic for such very intense extremes. Nevertheless, as we state in the discussion, even though the absolute values of the IC and the PR are uncertain, there is no doubt in this case that the event is attributable. Also, the direct comparison of the PR between the analogues and the EVT methods is not straightforward a priori and it is not clear how they should relate one to each other, because one is conditional and the other is not. We do not think that there is an issue in the quality of the analogues found for this event, as their composite is very close to the actual event in this case (see Figure 1d).

[Figure]

**Figure R2: as Figure 6 first column while removing the event itself from the fit.**

I ask the authors to revise or more fully justify these more implausible findings I've mentioned. It might aid interpretation to actually plot ~4 analogues (say the 15th, 30th, 45th and 60th closest analogues) for each case in the appendix, to see if a mismatch of meteorology between the event and analogues does indeed explain this.

**We thank the reviewer for this suggestion. However, showing just a few analogues may not be very informative of their full distribution. In the updated version of the article we rather added a figure (Figure 1def and L197-203) showing the average of the Z500 field and the metric field (temperature, wind and precipitation) for the analogues found with our method. Except maybe for the precipitation event, the synoptic mean field is close to the one of the event itself.**

**Minor/typographical comments**

1. Line 12: what is a 'climatological event'? A little unclear to me. Also 'negatively affect' reads more naturally than 'affect negatively'

**The 2021 IPCC report defines a climatological extreme event as "a pattern of extreme weather persisting for some time". We agree with the reviewer that this definition is**

**somewhat arbitrary but we prefer to keep it here to emphasize the possible different time scales involved. The rest of the sentence was corrected as suggested.**

    2. Line 15: 'which aim is to'

**This has been corrected.**

    3. Line 31: Given 'very low' doesn't have a well defined quantitative meaning, I'd just say 'estimating low probabilities'

**It could be possible to define a difference between low probability events (events for which few previous cases exist) and very low probability events (unseen events), but we agree with the reviewer that this may be confusing and we have modified the text as suggested.**

    4. Line 48: 'These methods both condition the ...'

**This has been corrected as suggested.**

    5. Line 57: I think by this point in the introduction it would be good to explicitly state the assumption of your work that changes in the synoptic dynamics themselves due to climate change are negligible/ should be treated separately. The conditional attribution is informative but by definition only a partial attribution to thermodynamic changes. As far as I can see this is not written in the introduction, and should be clearly emphasised.

**The following sentence has been added in the introduction as suggested: L55 "The conditional attribution thus separates the thermodynamical and dynamical changes due to climate change and addresses only the former."**

    6. Line 62: 'This method is used, for example, by the...'

**This has been corrected.**

    7. Line 68: 'Which lead to exceptional heat in...'

**This has been corrected.**

    8. Line 69: 'Western of Europe'

**This has been corrected.**

    9. Line 71: 'precipitations' (and several other times in the document)

**This has been corrected.**

    10. Line 81: Can you motivate why you take a 5 day mean for precip? I presume to focus on the synoptic driving.

The use of 5 day mean for precipitations is indeed intended to focus on the synoptic driving while daily precipitations risk being too variable to identify a signal using imperfect analogs.

The following sentence was added in the text: L81 "Using a 5-day rolling mean for precipitation allows us to focus on the synoptic driving rather than day-to-day variability."

11. Line 86: Even 1 degree is a fairly strong constraint on shifts. It might be interesting to see results for a 2.5/3 deg box average centred on the three points examined in figure 3.

**The results for a 3 degrees box centered on the three grid points in Figure 3 are presented in Figure R3 below. They are very similar to the ones of Figure 3.**

[Figure]

**Figure R3: as in Figure 3 but using a 3° by 3° box average centered on the grid points rather than the value at the grid point.**

12. Line 103: Clearer to say "thickening due to warming of the atmosphere".

**This has been changed as suggested.**

> 13. Line 111: Its not totally clear in what space you compute the analogues. The full high-dimensional space of gridpoint values within each region? That's giving you a O(1000) dimensional space. Given that distance vectors in high-dimensional space tend towards equal length (https://link.springer.com/chapter/10.1007/3-540-44503-X_27) which may lead to poor ability to detect analogues, did you consider computing analogues in an O(10-100) dimensional PCA space?

**We indeed compute analogues in the full O(1000) dimensional space of the Z500 field over the region considered, as in many other papers using analogues (e.g. Faranda et al. 2024). We understand the concern of the reviewer with regards to the ability of the method to detect good analogues in this high dimensional space, however in this space the actual number of degrees of freedom is much smaller than O(1000) because a lot of grid points are highly correlated. Tests made using a projection in a lower dimensional space, such as one obtained using a PCA, did not lead to major changes in the analogues found (see also Yiou (2014) and Jezequel et al. (2018)). For simplicity we therefore preferred using the original space.**

> 14. Line 127: impossible from the thermodynamic perspective, but this could be tackled by considering how synoptic conditions are being impacted by climate change.

**The conditional approach presented here - and the other methods cited in introduction also - is indeed close to a "thermodynamic" perspective for which we assume that the synoptic conditions have not changed and we project the event in the world with or without climate change. From the unconditional perspective, whether or not the dynamics of the yearly maximum events are different from the event studied is irrelevant as one only compares the intensity levels reached. We are not sure to understand what the reviewer refers to when they mentions "considering how synoptic conditions are being impacted by climate change", which is likely something difficult to assess with past data only (see also L57-58).**

> 15. Line 136: What is the motivation for using surface temperature for something like precip, given condensation happens aloft? If RMST is used just for simplicity then maybe make a comment about this somewhere.

**Here we do not really use surface temperature per se as a covariate - at least not local surface temperature - but RMST as a measure of the non stationarity of the climate system. This choice is standard in the attribution literature and we follow this practice here.**

> 16. Line 199: Worst-> worse

**This has been corrected.**

17. Fig 3abc: You should state in the figure caption that the 'detrended' data is shifted to the 2020 RMST value. It makes sense for your application, but isn't a standard detrending, where you'd expect to end up with zero mean displacement of the data.

**This has been added as suggested.**

18. Line 263: The intensity change your report does not correspond to what you show in fig 3i.

**This has been corrected.**

19. The captions for figures B5 and B6 are the same.

**This has been corrected.**

20. Line 345: missing reference.

**This has been corrected.**

**References**

**Jézéquel, A., Yiou, P., & Radanovics, S. (2018). Role of circulation in European heatwaves using flow analogues. *Climate dynamics*, *50*(3), 1145-1159.**

**Yiou, P. (2014). AnaWEGE: a weather generator based on analogues of atmospheric circulation. *Geoscientific Model Development*, *7*(2), 531-543.**

**Faranda, D., Vrac, M., Yiou, P., Jézéquel, A., & Thao, S. (2020). Changes in future synoptic circulation patterns: consequences for extreme event attribution. *Geophysical Research Letters*, *47*(15), e2020GL088002.**

**Platzer, P., Yiou, P., Naveau, P., Filipot, J. F., Thiébaut, M., & Tandeo, P. (2021). Probability distributions for analog-to-target distances. *Journal of the Atmospheric Sciences*, *78*(10), 3317-3335.**